# Lameness in Pregnant Sows Alters Placental Stress Response

**DOI:** 10.3390/ani13111722

**Published:** 2023-05-23

**Authors:** Marisol Parada Sarmiento, Lydia Lanzoni, Leandro Sabei, Matteo Chincarini, Rupert Palme, Adroaldo José Zanella, Giorgio Vignola

**Affiliations:** 1Center for Comparative Studies in Sustainability, Health and Welfare, Department of Preventive Veterinary Medicine and Animal Health, School of Veterinary Medicine and Animal Science, University of São Paulo, Pirassununga CEP 13635-900, SP, Brazil; sabei_le@alumni.usp.br; 2Department of Veterinary Medicine, University of Teramo, Loc. Piano d’Accio, 64100 Teramo, Italy; llanzoni@unite.it (L.L.); mchincarini@unite.it (M.C.); gvignola@unite.it (G.V.); 3Unit of Physiology, Pathophysiology and Experimental Endocrinology, Department of Biomedical Sciences, University of Veterinary Medicine, Veterinärpl. 1, 1210 Wien, Austria; rupert.palme@vetmeduni.ac.at

**Keywords:** fetal programming, glucocorticoids, pain, pig, welfare

## Abstract

**Simple Summary:**

Lameness is a painful and common condition in the pig production systems that makes locomotion difficult, negatively impacting sow welfare and profitability. This study demonstrates that the placenta of gestating lame sows was less efficient in inactivating stress hormones, an important mechanism to protect fetuses from the potentially damaging effect of maternal glucocorticoids. Our findings suggest that greater attention should be paid to the sows’ lameness because their welfare state may also affect their offspring’s welfare.

**Abstract:**

Pregnant sows from commercial pig farms may experience painful states, such as lameness, an essential indicator in assessing sow welfare. We investigated the effect of lameness during the last third of pregnancy on reproductive performance and placental glucocorticoid concentrations in sows. Periodic locomotion assessments were carried out on two commercial pig farms using a validated 0–5 scoring system (from 0: normal locomotion; to 5: a downer animal). Sows from both farms (N = 511) were grouped based on their average locomotion scores. On Farm 1, 30 sows were selected and grouped as either Not Lame (NL = 16; $\bar{X}$ = 0–1) or Lame (L = 14; $\bar{X}$ > 1). On Farm 2, 39 sows were selected and grouped as either Not Lame (G1 = 12; $\bar{X}$ = 0–1), Moderately Lame (G2 = 13; $\bar{X}$ = 1.1–2), or Severely Lame (G3 = 14; $\bar{X}$ ≥ 2.1). Reproductive data (gestation length, litter weight, average piglet weight, litter size, and the number of piglets born alive/mummified/stillborn) were recorded on both farms. Moreover, on Farm 2, piglet intrauterine growth restriction score and the number of piglets dead during the first week were also recorded, and placenta samples were collected to determine their cortisol/cortisone concentrations. A linear mixed model was used to analyze the data. The proportion of lameness in pregnant sows (N = 511) was >40%, and the gestation length tended to decrease with the presence of lameness (*p* < 0.1) in both farms. G2 sows had a higher placental cortisol/cortisone ratio than G1 and G3 sows (*p* < 0.01). In conclusion, lameness was high in the sows assessed, which may decrease sow gestation length and reduce placental efficiency in protecting the offspring from the sows’ stress response.

## 1. Introduction

Lameness in pregnant sows is a common and painful condition, being one of the primary reasons for sow removal during the early peripartum period, impacting not only the economic performance of the farm but also the welfare of the sow and her offspring [1,2,3,4]. Lameness affects farm productivity by reducing sow longevity, and it has been reported that about 9% of sow culling is due to lameness or foot lesions [2,5]. Additionally, as a painful and, therefore, stressful condition, lameness can alter the cortisol levels of pregnant sows, possibly affecting their offspring’s in utero development [6,7].

Sow lameness is multifactorial in origin, with many different factors contributing to the high prevalence of lameness on commercial pig farms [8]. For example, a large herd size and injuries due to post-mixing aggression have been associated with lameness in sows [8]. Furthermore, abrasive flooring and nutritional factors, such as mineral imbalances, can be detrimental to the bone, articular cartilage, and hoof quality [9].

The economic consequences of lameness have been investigated extensively, with both premature sow death and increased voluntary culling of lame sows associated with significant economic losses [10,11]. Farm economic losses caused by premature disposal were estimated by Dijkhuizen et al. [12] to be 16% of the total farm income. In addition, a high incidence of lameness can lead to lower payments from the slaughterhouse and/or increased carcass condemnations [11].

Lameness is a painful condition and has been recognized as one of the best animal-based indicators of welfare in pigs [13]. The pain experienced by the lame animal activates the sympathetic/autonomic nervous system and the hypothalamic-pituitary-adrenal axis (HPA) [7]. Physiological changes associated with HPA activation are suitable biomarkers to assess animal pain, and glucocorticoids are the main stress hormones used as pain and stress indicators [6,7]. During pregnancy, glucocorticoids are essential for the development and maturation of internal organs to prepare the fetus for the extrauterine environment. However, in high concentrations, when produced in response to stress or inflammation, glucocorticoids have negative consequences on fetal programming [14].

The placenta has mechanisms to protect the fetus from maternal glucocorticoids, such as the release of 11 beta-hydroxysteroid dehydrogenase type 2 [15]. This enzyme is responsible for the inactivation of maternal glucocorticoids, transforming cortisol (hydrocortisone) into cortisone, thus reducing prenatal fetal exposure to cortisol and mitigating its negative effects on the offspring [15]. Sows that receive oral administration of hydrocortisone acetate, a compound that increases the concentration of plasma cortisol, have a shorter gestation length (88.2 ± 11.9 days) [16].

Lameness in pregnant sows modifies the body weight and aggressive behavior of their offspring at weaning [1], indicating that sow welfare during pregnancy may alter the offspring’s physiology, behavior, and performance.

The high prevalence of lameness in commercial pig farms, and its known adverse welfare effects, support the importance of further researching its consequences [17,18,19,20]. The objectives of the present research were to identify the effect of lameness in sows during the last third of pregnancy on their reproductive performance (gestation length, total litter size, number of born alive/mummified/stillborn piglets, piglet intrauterine growth restriction, litter weight, average piglet weight, and piglet mortality during the first week post-partum) and placental glucocorticoid concentrations. We hypothesized that sow lameness would have a negative effect on reproductive performance and result in a higher concentration of placental glucocorticoids.

## 2. Materials and Methods

Data collection was carried out on two farms in two countries—Brazil and Italy. Data from pregnant sows with different degrees of lameness were collected on a Brazilian pig farm in the state of Parana (November–January) and on an Italian pig farm in the region of Abruzzo (June–August). Data collection in Brazil was carried out a year before that in Italy. Both studies were conducted under similar weather conditions, with average temperatures of 24 °C in Parana (Brazil) and 25 °C in Abruzzo (Italy).

### 2.1. Animals, Handling, and Locomotion Assessment

On the Brazilian pig farm (Farm 1), pregnant sows (TOPGEN^®^ Landrace/Large white) were housed in group gestation pens that measured 6 × 4 m (2.7 m^2^ per sow) with walls 0.9 m high and with both solid (16 m^2^) and slatted (8 m^2^) concrete floor areas. Sows were fed a liquid gestation diet twice daily (1.8 kg per sow/day—06:00 h and 15:00 h) in group feed troughs (5 m long and 0.4 m wide). As per farm management, all sows were transferred to individual farrowing crates (2.6 × 1.6 m) one week before the expected farrowing date based on the insemination date. Births were assisted 24 h a day by farm technicians. During lactation, all sows were fed with a lactation diet four times per day (07:00 h, 11:00 h, 16:00 h, and 21:00 h), 8 kg per sow/day.

On the Italian pig farm (Farm 2), pregnant sows (TOPIGS^®^ TN70) were housed in group gestation pens that measured 6 × 7 m (2.8 m^2^ per sow), with both solid (9 m^2^) and slatted (33 m^2^) concrete floor areas. Sows were fed a dry gestation diet twice daily (1.3 kg/sow/time—07:00 h and 15:00 h). As per farm management, all sows were transferred to individual farrowing crates (2.4 × 1.6 m) one week before the expected farrowing date based on the insemination date. Births were only assisted during the day. During lactation, all sows were fed a lactation diet three times per day (07:00 h, noon, and 16:30 h) 1.0 kg/sow/time starting the day after farrowing, increasing 300 g daily until complete 2.6 kg/sow/time. On both farms, sows were grouped in gestation pens according to their expected farrowing date and size; water was supplied ad libitum by a nipple drinker.

On both farms, the same validated 0–5 locomotion scoring system was used by the same trained observer to assess sow lameness, from 0, for an animal that walks easily, to 5, for a downer animal (Ref. [21] sees Table 1 for details about each score). On both farms, the assessment was performed weekly in the sow group gestation pens during the last third of pregnancy. While the trained evaluator observed, an assistant guided the animal to walk on the pen solid floor area to assess the locomotion score. In both farms, sows with a locomotion score ≥ 3 were treated with 1.1 mg/kg of flunixin meglumine intramuscularly for three days, as requested by the ethics committee. Sows with a locomotion score of 5 were removed from the collective pen into an individual pen.

The study used locomotion data from 397 sows on Farm 1 and 114 sows on Farm 2, comprised of all sows with at least three locomotion assessments during the last third of pregnancy. On Farm 1, sows were divided into two groups according to their average locomotion scores: Not Lame (NL) sows with an average locomotion score between 0 and 1, and Lame (L) sows with an average locomotion score > 1. On the second farm, it was decided to include a moderately lame group to control for the possible effect of analgesics administered to sows with a locomotion score ≥ 3. Therefore, on Farm 2, the sows were divided into three groups: not lame (G1), sows with an average locomotion score between 0–1, moderately lame (G2) sows with an average score between 1.1–2, and severely lame (G3) sows with an average score ≥ 2.1 (see Figure 1 for details).

### 2.2. Experimental Data Collection

After farrowing, reproductive data were collected from a sample of sows that presented with consistent locomotion scores throughout the last third of pregnancy (a score equal to or close to the previous assessment) and their proximity to their expected farrowing date (determined based on the date of insemination and occurring within one month). This resulted in reproductive data from 30 sows on Farm 1 (NL = 16; L = 14) and 39 sows on Farm 2 (G1 = 12; G2 = 13; G3 = 14). Reproductive data acquired on both farms were: gestation length, litter weight, average piglet weight, piglet sex, total litter size, and the number of piglets born alive/mummified/stillborn.

#### Intrauterine Growth Restriction and Placental Cortisol and Cortisone—Farm 2

Additionally, due to the facilities available on Farm 2, it was possible to also record: piglet mortality during the first week postpartum, the individual physical characteristics of the piglets to establish an intrauterine growth restriction (IURG) score, and to collect placental samples subsequently used to measure cortisol and cortisone.

IURG was evaluated using the methodology described by Hales et al. [22], with piglets scored based on the visual presence of three head morphologies: (1) a steep, dolphin-like forehead; (2) bulging eyes; and (3) wrinkles perpendicular to the mouth. If none of the three morphologies were present, the piglet was defined as normal (score 1); if one or two of the morphologies were present, the piglet was defined as light IURG (score 2); and piglets with all three morphologies were defined as IURG (score 3).

To measure cortisol and cortisone, three random placentas were chosen per sow immediately after farrowing, with tissue samples collected in the same location from each placenta (see Figure 2 for details) and stored in plastic containers at −20 °C. The three tissue samples per sow were mixed and homogenized into a powder with a porcelain mortar and pestle, using liquid nitrogen to prevent them from defrosting. Afterward, samples were again stored at −20 °C. For glucocorticoid extraction, an aliquot of 0.1 g of each homogenized sample was weighed into a 1.5 mL microtube and mixed with 200 µL of ultrapure water in a hand vortex for 15 s. Then, 1 mL of ethyl acetate was added and remixed in a hand vortex for 15 s before being centrifuged at 4 °C at a speed of 4000× *g* for 20 min. The supernatant was then delicately removed and frozen in a 1.5 mL microtube at −20 °C. Finally, the microtubes were removed from the freezer, and 500 µL of the supernatant was collected into a new microtube and dried under a hood. Each dry microtube was resuspended in 500 µL of EIA buffer, shaken for 30 min, and 50 µL measured in duplicate in cortisol and cortisone enzyme immunoassays (EIAs). Details of the EIAs have been described previously [23,24,25].

Protein concentration was determined using an adaptation of the Bradford [26] technique. Placental tissue samples (0.1 g) were homogenized in a lysis buffer containing 150 mM NaCl and with a 1% protease inhibitor. Homogenization was carried out using a sonicator for nine cycles of 10-s pulse with 10-s gaps at 35% amplitude (range 10–100%) to completely lyse the tissues. The homogenates were centrifuged for 15 min at 14,000× *g* at 4 °C, and the resulting supernatants were assayed for protein concentration using a Bio-Rad Protein assay. Sample protein concentration was established against a standard curve generated using Bovine Serum Albumin. Each sample was analyzed in duplicate and read at 595 nm by a microplate reader to measure the optical density values.

Placental cortisol and cortisone concentrations were corrected using the placental protein concentration [25] as follows: Placental cortisol or cortisone corrected = cortisol or cortisone concentration [pg/µL]/placental protein [pg/µL]. The ratio between placental cortisol and cortisone levels was included in the statistical model.

### 2.3. Statistical Analysis

A linear model was used to test the effect of the sow lameness group (Farm 1: NL and L; Farm 2: G1, G2, and G3) on reproductive data and the ratio of placental glucocorticoids (for Farm 2). The model included the sow lameness group, sow parity, and the total litter size as fixed effects.

A mixed model was used to test the effect of the sow lameness group, sow parity, piglet sex, the total litter size, and piglet IURG score (for Farm 2) on individual piglet body weight. The sow was included as a random effect.

To analyze the effect of sow lameness on piglet IURG scores for Farm 2, a linear mixed model was used, with sow lameness group, sow parity, the total litter size, and piglet sex as fixed factors, and the sow as a random effect.

The method used for multiple comparisons and *p*-value adjustment was Tukey.

Results were considered significant when the *p*-value was <0.05 and a tendency when the *p*-value was between 0.05 and 0.1. All analyses were performed with R statistical software RStudio 2022.07.2+576 [27] using the packages readxl, tidyverse, rstatix, emmeans, easystats, and lmerTest [28,29,30,31,32,33].

## 3. Results

### 3.1. Locomotion and Reproductive Measures

The number and percentage of lame sows by locomotion group for both commercial pig farms are presented in Table 2.

Descriptive measures and main results from Farms 1 and 2 are presented in Table 3 and Table 4. On Farm 1, lame sows tended to have a lower gestation length than not lame sows (NL = 115.6; L = 114.5 days; *p* = 0.06), while in Farm 2, there was a trend for locomotion group effect, with no difference between groups (G1 = 116.1; G2 = 114.6; G3 = 115.3 days; *p* = 0.06).

On Farm 1 and 2, no significant differences existed between the locomotion groups on any of the reproductive performance measures.

When analyzing the effects of the sow locomotion group, piglet sex, and piglet IURG score on piglet body weight, we found an association between piglet IURG score and birth weight. Piglets with an IURG score of 1 were heavier than piglets with IURG scores of 2 or 3 (*p* < 0.0001), whilst piglets with a score of 2 were heavier than piglets with a score of 3 (*p* < 0.0001). There was no effect of the sow locomotion group on the IURG score. No other significant effects or differences were found in the reproductive or IURG data.

### 3.2. Placental Measures

There was a significant effect of the sow lameness score group on the placental cortisol to cortisone ratio (*p* = 0.03; see Figure 3), with the ratio higher for G2 sows than for G1 and G3 sows (*p* < 0.05). No differences or effects were found when using cortisol or cortisone concentrations alone.

## 4. Discussion

The proportion of lame sows among those assessed during the last third of pregnancy was disturbingly high in both farms, especially for scores ranging from moderate to severe, being higher than reported by Pluym et al. [18] (9.7%) but similar to data reported by Kramer [17] and Sobestiansky [20] (65.2% and 50%, respectively). Despite the discrepancy between the lameness prevalence found in the literature, the values found in the present study suggest that a high proportion of pregnant sows on commercial pig farms could be experiencing pain. Lameness is considered a crucial problem because it not only affects the profitability of the farms but can also severely compromise the welfare of the animals [4].

The sow gestation lengths found in the current study are within the physiological range for the species [34]. However, lameness as a stressful condition has the potential to alter the gestation length of pregnant sows. Following Kattesh et al. [35] and Kranendonk (2005), the length of gestation in the current study tended to be shorter in sows exposed to a stressor (lameness) during pregnancy on Farm 1. Kattesh et al. [35] found that sows exposed to stressors during pregnancy have a reduced gestation length from 115.3 to 112.3 days, while Kranendonk [16] found a reduced gestation length in sows that had higher plasma cortisol concentrations (due to the administration of hydrocortisone acetate). On the contrary, Ashworth et al. [36] found no effect of social stress on sow gestation length. High cortisol concentrations, as a consequence of stressful conditions, such as lameness, could perform an essential role in decreasing the gestation length. It has been reported that high concentrations of endogenous and exogenous glucocorticoids cause preterm parturition [16]. Therefore, we suggest including cortisol measures over time in future studies involving lameness in pregnant sows and its reproductive consequences.

From a physiological perspective, it has been reported that glucocorticoids during pregnancy participate as catabolic agents in growth processes, restraining fetal growth and tissue development [37]. G1 sows were more efficient in inactivating placental cortisol into cortisone than G2 sows, probably protecting their fetuses from increased exposure to cortisol. However, no difference was observed in piglet IURG scores between sow lameness score groups. The enzyme 11 beta-hydroxysteroid dehydrogenase type 2 [15] is essential for the inactivation of placental cortisol into cortisone; therefore, we suggest measuring it in future studies. Locomotion score during early and mid-gestation is unknown because it was not assessed in the present study. However, the manifestation of lameness could be reduced in these stages since the sows could be lighter compared to the pre-farrowing period [2].

Whilst clear differences were found in placental glucocorticoid ratios when comparing the not lame and moderately lame sow groups, no differences were found when comparing the not lame and severely lame sow groups. These results may be related to the fact that sows with severe lameness were treated with analgesics (flunixin meglumine), possibly decreasing their pain and stress responses [38,39,40].

## 5. Conclusions

In conclusion, sow lameness during the last third of pregnancy is a relevant welfare issue because a high proportion of the sows monitored in the present study were found to suffer from this stressful and painful condition. Moreover, we found that sow lameness reduced offspring protection from cortisol due to decreasing placental efficiency in inactivating cortisol to cortisone. Moderate lameness was enough to modify physiological parameters in the placenta and tended to decrease gestation length. Sow lameness is often only recognized when it becomes severe or chronic, reducing the possibility of early intervention [41], therefore, moderate or subclinical lameness could represent a significant challenge for the swine industry. Educational interventions in lameness prevention and recognition of moderate sow lameness could be targeted at swine farmers to improve animal welfare and the sector’s productivity. Further studies may contribute more information about the effects of sow lameness during pregnancy throughout the life trajectory of the offspring, as reported in a previous study [1].

## Figures and Tables

**Figure 1 animals-13-01722-f001:**
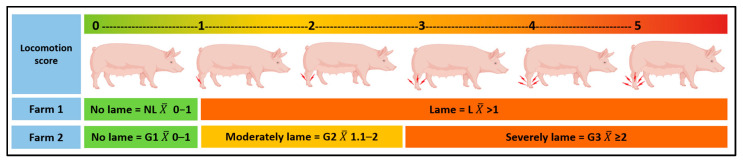
Grouping of sows on each farm based on their average locomotion assessment. A validated 0–5 scoring system was used from 0 for normal locomotion to 5 for a downer animal [21]. On Farm 1: NL is a Not Lame sow (average of assessments: 0–1), and L is a Lame sow (average of assessments > 1). On Farm 2: G1 is a not-lame sow (average of assessments: 0–1), G2 a sow with moderate lameness (average of assessments: 1.1–2), and G3, a sow with severe lameness (average of assessments: ≥ 2).

**Figure 2 animals-13-01722-f002:**
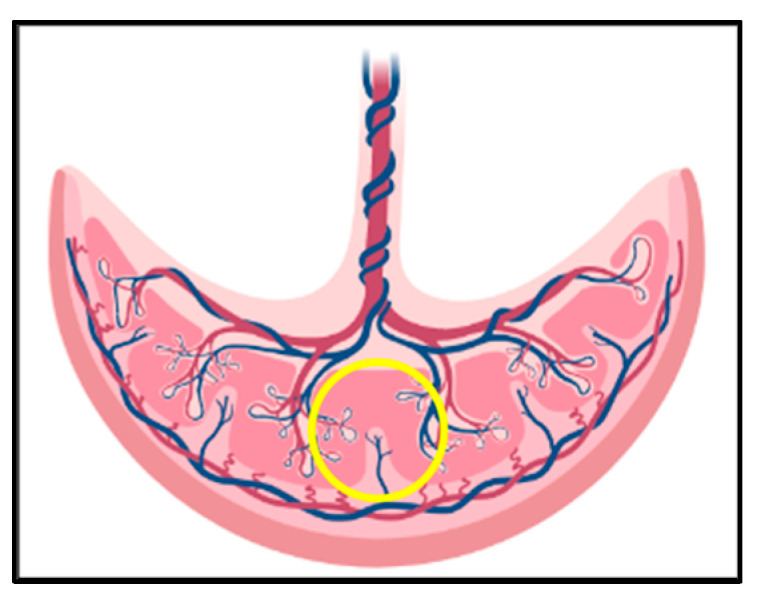
Placental sampling region. The yellow circle illustrates the region used to collect placenta samples from sows on Farm 2.

**Figure 3 animals-13-01722-f003:**
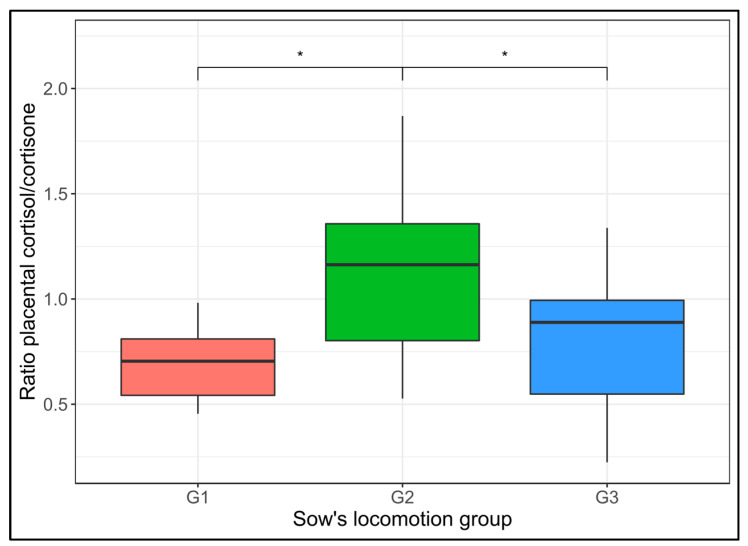
Placental cortisol and cortisone ratios grouped by sow locomotion score group from Farm 2. G1: a not lame sow (average of assessments: 0–1); G2: a sow with moderate lameness (average of assessments: 1.1–2); and G3: a sow with severe lameness. *: significant (*p* < 0.05).

**Table 1 animals-13-01722-t001:** Locomotion score system used to assess lameness in pregnant sows. Reprinted with permission from Ref. [21]. License Number 5554171425700, 22 May 2023, Licensed Content Publisher: Cambridge University Press, Licensed Content Publication: Animal Welfare.

Score	Label	Description
0	Normal	Even strides, rear end sways slightly while walking, pig is able to accelerate and change direction rapidly. Stands normally.
1	Stiff	Abnormal stride length, movements no longer fluent, pig appears stiff. Pig still able to accelerate and change direction. Stands normally.
2	Slight lameness	Shortened stride, lameness detected, swagger of rear end while walking, no hindrance in pig’s agility. Uneven posture while standing.
3	Lame	Pigs slow to get up (may dog sit), shortened stride, minimum weight-bearing on affected limb (standing on toes), swagger of rear end while walking. May still trot and gallop.
4	Limping	Pig reluctant to get up, holds limb off floor while standing, avoids placing affected limb on the floor while moving.
5	Downer	Pig unresponsive: does not move and struggles to stand when encouraged to do so.

**Table 2 animals-13-01722-t002:** Number and percentage of lame sows by locomotion group for both commercial pig farms.

	Groups	Number of Sows	Proportion (%)
Farm 1	NL	224	56.4
	L	173	43.6
	Total	397	-
Farm 2	G1	31	27.2
	G2	59	51.8
	G3	24	21.1
	Total	114	-

Abbreviations: On Farm 1: NL is a Not Lame sow (average of assessments: 0–1), and L is a Lame sow (average of assessments > 1). On Farm 2: G1 is a not-lame sow (average of assessments: 0–1), G2 a sow with moderate lameness (average of assessments: 1.1–2), and G3, a sow with severe lameness (average of assessments: ≥2).

**Table 3 animals-13-01722-t003:** Description and results of reproductive measures obtained on Farm 1 divided by sow locomotion group.

Variable	No Lame N = 16		Lame N = 14		Effect *p*-Value		
	Mean	SEM	Mean	SEM	Locomotion Group	Parity	Litter Size
Gestation length (days)	115.6	0.47	114.5	0.24	0.06 ^t^	0.68	0.18
Average litter weight (kg)	18.9	1.12	18.8	0.97	0.90	0.01	<0.01
Average piglet weight (kg)	1.60	0.09	1.50	0.07	0.16	0.18	<0.01
Number of male piglets	6.00	0.69	6.90	0.65	0.17	<0.05	<0.01
Number of female piglets	6.70	0.59	6.20	0.65	0.44	0.60	<0.01
Total number of piglets	13.6	1.01	14.2	0.97	0.65	0.57	-
Number of live piglets	12.7	0.95	13.1	0.89	0.32	<0.01	<0.01
Number of mummified piglets	0.20	0.19	0.30	0.16	0.70	0.74	<0.05
Number of stillborn piglets	0.70	0.34	0.80	0.25	0.82	0.54	0.36

Abbreviations: No-lame sow (average of assessments: 0–1), Lame sow (average of assessments > 1). Mean: average; SEM: standard error of the mean. ^t^: tendency (*p*-value between 0.05 and 0.1).

**Table 4 animals-13-01722-t004:** Description and results of reproductive and placental measures obtained on Farm 2 divided by sow locomotion group.

Variable	G1 N = 12		G2 N = 13		G3 N = 14		Effect *p*-Value			Tukey Adjusted *p*-Value		
	Mean	SEM	Mean	SEM	Mean	SEM	Locomotion Group	Parity	Litter Size	G1–G2	G1–G3	G2–G3
Gestation length (days)	116.1	0.33	114.6	0.49	115.3	0.40	0.06 ^t^	0.29	0.95	0.14	0.93	0.29
Average litter weight (kg)	18.1	1.22	18.8	1.27	18.7	0.94	0.92	0.91	<0.05	0.94	0.94	0.77
Average piglet weight (kg)	1.50	0.08	1.40	0.06	1.30	0.07	0.19	0.66	<0.001	0.87	0.59	0.85
Number of male piglets	6.20	0.64	7.10	0.75	7.20	0.66	0.49	0.41	<0.01	0.78	0.84	0.99
Number of female piglets	6.10	0.87	6.40	0.87	6.80	0.54	0.79	0.46	0.02	0.98	0.99	0.95
Total number of piglets	15.0	0.75	15.8	1.10	16.4	0.88	0.60	0.82	-	0.98	0.79	0.87
Number of live piglets	12.7	0.87	14.2	1.01	14.3	0.58	0.02	0.14	<0.001	0.25	0.99	0.21
Number of mummified piglets	0.60	0.26	0.30	0.13	0.90	0.38	0.22	0.08 ^t^	0.08 ^t^	0.85	0.47	0.19
Number of stillborn piglets	1.70	0.45	1.30	0.47	1.20	0.32	0.60	<0.01	0.11	0.24	0.68	0.71
Number of piglets dead in the 1st week	1.20	0.28	2.50	0.67	2.20	0.74	0.25	0.44	<0.001	0.22	0.52	0.83
Placental cortisol/cortisone ratio	0.77	0.08	1.17	0.15	0.78	0.09	0.03	0.47	0.16	0.05	0.93	0.02

Abbreviations: G1 is a no-lame sow (average of assessments: 0–1), G2 is a sow with moderate lameness (average of assessments: 1.1–2), and G3 is a sow with severe lameness (average of assessments: ≥2). Mean: average; SEM: standard error of the mean; ^t^: tendency (*p*-value between 0.05 and 0.1).

## Data Availability

Data supporting the findings of this study can be found in Appendix A.

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
