# Peer review of "Lameness in Pregnant Sows Alters Placental Stress Response"

_animals, 2023, doi:10.3390/ani13111722_

Round 1

Reviewer 1 Report

95: Presumably the timing was directed by the seasons in opposite hemispheres?  Perhaps indicate this, with months mentioned, and compare average summer temperatures?

129: Indicate the treatment strategy with analgesics used. Was only flunixin used?

143: Explain what is meant by "consistent" and "proximity to their expected farrowing date".

262: Some analysis of the lameness scores across the three measures could possibly help explain this.  What were the individual scores? Were there increases with the progressing pregnancy and weight gain?

Author Response

Dear Reviewer 1,

Thank you very much for your time dedicated to reviewing our manuscript. Addressing your comments certainly will improve our document.

Specific answers to each of your comments and supporting files are attached.

Reviewer 2 Report

Ethical concerns:
1. Treatment of downer (lameness category 5) sows in group housing. Access to feed and water can be severely impacted.
2. Use of flunixin meglumine in sows. The product information specifies that the product is not for use in breeding swine since the effects on reproduction have not been studied.  Only a prescription from a veterinarian can override the product label. This applies in North America; I am not sure of the regulations in Brazil and Italy.

Methods:
1. Overall N is low. What size of effect can this experiment detect (what is the power of the experiment) ?
2. Not enough information is provided to reproduce the experiment. Floor type is reported for only one of the two farms. No information is reported on environmental factors, such as temperature. No information is provided on the composition or nutritional values of the feeds used, or the amount of feed provided. No information is provided on health status. Were sows grouped by size, by insemination date, or some combination of these?

Statistical analysis:
1. Ordered variables, including parity and litter size, should be treated as fixed rather than random effects.
2. With a mixed model, variables included in the original model may or may not be found to have significant effects. These should be reported.

Conclusions:
1. Parity can affect both lameness and gestation length; for example, see Yang KY, Jeon JH, Kwon KS, Choi HC, Kim JB, Lee JY. Effect of different parities on reproductive performance, birth intervals, and tail behavior in sows. J Anim Sci Technol. 2019 May;61(3):147-153. doi: 10.5187/jast.2019.61.3.147. Epub 2019 May 31. PMID: 31333871; PMCID: PMC6582923. The statistical model used cannot determine how much of the effect is due to parity.
2. Although differences were found in cortisol/cortisone ratios, there were no differences in cortisol among the lameness groups. This does not support the conclusion that sow lameness decreased offspring protection from cortisol.

Language concerns are highlighted in the text.

Author Response

Dear Reviewer 2,

Thank you very much for your time dedicated to reviewing our manuscript. Addressing your comments certainly will improve our document.

Specific answers to each of your comments and supporting files are attached.

Reviewer 3 Report

Overall, the data presented in this paper are highly interesting and provide valuable insights for the research community. The manuscript describes a straightforward and easily understandable experiment with a clear objective, and the relevant literature is included to justify the study. Although the Material and Methods section provides information on the methodology, further details should be added to facilitate the replication of the experiment in future studies.

The Results and Discussion sections are well written, however, I suggest that some minor improvements could be made in the latter regarding the main findings of the study. For example, the authors should explain the link between lameness (stress) and shortened gestation lengths. Is there a known physiological explanation or a hypothesis that could explain this relationship? 

Here are some specific comments on the manuscript:

Abstract

L38-39: Please verify and clarify the sentence …lameness was high among the assessed sows…

Keywords: Please verify the presence of number in the keywords.

Introduction:

L59: Rejections or condemnations?

L74-75: Please, provide the range of shortened gestation lengths.

Materials and Methods

L116: Did the observers receive training together (calibration) before starting observations in the field?

L116-121: Please explain how lameness was assessed. For example, were the sows made to walk a known distance? Was the observer inside the pens?

L127: Why were the groups not divided using the same criteria (mild, moderate, and severe)? Was this decision made due to the incidence of sows showing mild to moderate lameness in the Brazilian farm?

L141-147: Were the reproductive data collected by the study group or using the farm records?

L163- 165: Is it correct that placentas were only collected from Farm 2 (Italy)? If so, please clarify this in the current section.

Results

L227: Could the group division, which assembled mild, moderate, and severe lameness, have influenced the results?

Discussion

L262-264: Especially for scores ranging from moderate to severe.

L270-272: How can the authors explain the link between lameness (stress) and shortened gestation lengths? Is there a known physiological explanation or a hypothesis that could explain this relationship? It is suggested that this information be included in the discussion section.

L289-296: Is it possible that moderate lameness in group G2 is more stressful for sows than mild and severe lameness, as found in groups 1 and 3? Or is this result because severe lame (G3) was treated compared to G2? It is suggested that this information be included in the discussion section.

Regards, 

Author Response

Dear Reviewer 3,

Thank you very much for your time dedicated to reviewing our manuscript. Addressing your comments certainly will improve our document.

Specific answers to each of your comments and supporting files are attached.

Round 2

Reviewer 2 Report

Ethical concerns have been addressed. Description of methods and statistics is much improved.

No problems understanding the content, but some polishing of language is needed.